# On the creation of narrow AI:
# hierarchy and nonlocality of neural network skills

**Eric J. Michaud**[1,3*]    **Asher Parker-Sartori**[2]    **Max Tegmark**[1,3]

[1] Department of Physics, Massachusetts Institute of Technology
[2] Department of EECS, Massachusetts Institute of Technology
[3] The NSF AI Institute for Artificial Intelligence and Fundamental Interactions

## Abstract

We study the problem of creating strong, yet narrow, AI systems. While recent AI progress has been driven by the training of large general-purpose foundation models, the creation of smaller models specialized for narrow domains could be valuable for both efficiency and safety. In this work, we explore two challenges involved in creating narrow AI systems, having to do with basic properties of how neural networks learn and structure their representations. The first challenge regards when it is possible to train narrow models from scratch. Through experiments on a synthetic task, we find that it is sometimes necessary to train networks on a wide distribution of data to learn certain narrow skills within that distribution. This effect arises when skills depend on each other hierarchically, and training on a broad distribution introduces a curriculum which substantially accelerates learning. The second challenge regards how to transfer particular skills from large general models into small specialized models. We find that model skills are often not perfectly localized to a particular set of prunable components. However, we find that methods based on pruning can still outperform distillation. We investigate the use of a regularization objective to align desired skills with prunable components while unlearning unnecessary skills.

## 1   Introduction

Today, the most competent AI systems in *any* particular domain are general systems that are relatively competent in *every* domain. The best models at math and coding are also broadly knowledgeable about a very diverse array of topics, from Roman history to home cooking recipes to medical diagnostics. And when domain-specific models are created today, they are typically general foundation models fine-tuned on a particular task, rather than new models trained from scratch [1, 2], though with some notable exceptions [3]. This state of affairs is convenient and powerful, since a single general model can be used for a variety of applications [4].

However, generality has downsides for both efficiency and safety. For instance, AI systems used as coding assistants possess a large amount of knowledge which is never needed in those applications. Instead, we might like to use smaller, specialized networks which preserve the coding knowledge of general systems without the same breadth of irrelevant knowledge. Narrow systems may also pose fewer safety risks than general systems. For instance, narrow systems may have fewer dangerous capabilities that pose CBRN risks [5], be easier to understand mechanistically [6, 7], or be easier to verify properties of [8–10]. More imaginatively, for systems to operate autonomously in the world requires a large breadth of skills, and an ecosystem of narrow "tool AI" systems may therefore reduce loss-of-control risks and better support human agency over the long term [11].

In this work, we investigate some basic questions about how neural networks *learn* and *implement* skills that are relevant to the problem of creating narrow AI systems. As summarized in Figure 1,

---

*ericjm@mit.edu

39th Conference on Neural Information Processing Systems (NeurIPS 2025).

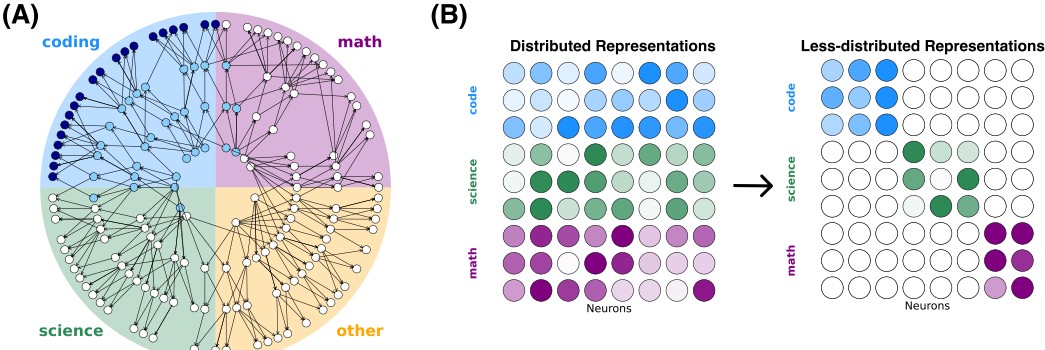

Figure 1: We study two challenges to making strong, narrow-purpose AI models. **(A)**: Data may have *hierarchical* structure. If skills have a hierarchical dependence, where some skills are only learnable after more primitive skills are learned first, then it sometimes may be necessary to train on a broad distribution of data to learn certain narrow skills within that distribution. These dynamics may mean that general-purpose models must be trained to achieve good performance on some domains. **(B)**: Model features are *distributed*. By default, skills may not be localizable to a particular set of model components (e.g. neurons). In this case, pruning of model components won't precisely retain wanted skills and remove unwanted skills from models. We explore methods for aligning the model features relevant to particular domains with a smaller subset of model components while unlearning others.

we focus on two main themes. First, we consider the question of when it is possible to train well-performing neural networks from scratch on a narrow data distribution. Through experiments on a novel synthetic task with *hierarchical structure*, we find that it can be necessary to train networks on a broad distribution to efficiently learn narrow tasks within that distribution. These results contribute to a growing literature on how task structure influences neural network learning dynamics [12–15], but with special relevance to the problem of creating narrow AI. Second, we consider the question of whether one can use pruning to turn broad networks into smaller narrow ones. We find that the *nonlocality* of network representations to prunable model components poses a challenge for this goal. While distributed representations have been extensively studied in the context of neural network interpretability [16] and in the classical connectionist literature [17, 18], we study how this property of neural network computation impacts pruning and unlearning [5] for creating narrow AI. Our specific contributions are as follows:

- We describe a synthetic task, *compositional multitask sparse parity* (CMSP), extending the multitask sparse parity task of Michaud et al. [19]. We find that networks trained on this task exhibit extremely strong curriculum learning effects, where it is necessary to train on a broad distribution of tasks in order to learn certain other tasks.

- We study pruning and unlearning on our networks trained on CMSP. We observe that tasks are often distributed and distinct subtasks are entangled, making pruning an imperfect strategy for "narrowing" the breadth of network skills. However, we find that a simple group-sparsity regularization objective can be used to sparsify networks and unlearn skills.

- We perform an empirical comparison of methods for creating narrow systems on MNIST and in LLMs. We tentatively find that methods based on pruning outperform distillation and training networks from scratch for the creation of smaller, more narrow systems.[2]

This work is organized as follows. In Section 2, we briefly describe methods. In Section 3 we perform a detailed case study of training and pruning on the CMSP task. We define the task (Section 3.1), study network training dynamics on it (Section 3.2), find distributed representations in these networks (Section 3.3), and use a regularization method for pruning and unlearning in these networks (Section 3.4). We then compare various methods for creating narrow models, studying MNIST in Section 4 and language models in Section 5. We discuss related work in Section 6 and conclude in Section 7.

---

[2]Code for the experiments in this paper can be found at: `https://github.com/ejmichaud/narrow` .

## 2 Methods

**Pruning:** We aim to preserve the performance of a model $f(\cdot;\theta)$ on a distribution $\mathcal{D}_N$ while pruning model components. Let $g$ be a collection of parameter indices $i$ corresponding to prunable components of the model (e.g. the in-weights and out-weights of a neuron), and let $G$ denote the collection of all such groups. After ablating a specific $g$, we denote the new parameters $\theta_g^*$. To perform pruning, for each $g \in G$, we compute an *ablation score* $s_g = \left| \mathbb{E}_{(x,y)\in\mathcal{D}_N} \left[ L(f(x;\theta),y) - L(f(x;\theta_g^*),y) \right] \right|$, the absolute change in the model's expected loss $L$ after pruning $g$. We sort groups by their ablation score and prune greedily to the desired sparsity from lowest to highest ablation score. Where feasible, we estimate $s_g$ empirically by manually ablating the group across many samples from $\mathcal{D}_N$. When this is computationally intractable, we instead use a linear estimate which we refer to as an *attribution score* after [20, 21]: $\hat{s}_g = \left| \sum_{i\in g} \frac{\partial L}{\partial \theta_i}(-\theta_i) \right| = \left| \frac{\partial L}{\partial \theta} \cdot (\theta_g^* - \theta) \right|$, where $\partial L/\partial \theta$ is the model's gradient on the distribution $\mathcal{D}_N$, which can be computed once and reused for all $g \in G$.

**Regularization:** We also experiment with making networks more prunable by performing additional training with a "group lasso" regularization penalty on the model weights [22–25]. The group lasso penalty $R$ is the L1 norm of the L2 norms of each parameter group: $R(\theta) = \sum_{g\in G} \sqrt{\sum_{i\in g} \theta_i^2}$. This penalty incentivizes the weights to become sparse at the level of entire groups $g$. When we perform "group lasso training" on a distribution $\mathcal{D}_N$, we minimize the loss $\mathbb{E}_{(x,y)\sim\mathcal{D}_N}[L(f(x;\theta),y) + \lambda R(\theta)]$.

**Distillation:** We also explore distilling knowledge from a teacher model using the standard algorithm employed in [26], minimizing the KL divergence between student and teacher output distributions.

## 3 Case study: compositional multitask sparse parity

In this section, we conduct a detailed study of both curriculum learning and pruning on simple synthetic task, which we call *compositional multitask sparse parity* (CMSP).

### 3.1 Defining compositional multitask sparse parity (CMSP)

The compositional multitask sparse parity (CMSP) task is a simple extension of the *sparse parity* task recently studied in [27] and the *multitask sparse parity* task studied in [19], described below.

Barak et al. [27] studied neural network training dynamics on the *sparse parity* (SP) task. This is a binary classification problem on binary strings $x$ of length $n$, where the label of a given sample $x$ is the parity of the bits at a subset $I$ of $k$ indices: $y = \oplus_{l=1}^{k} x[I_l]$. Strings $x$ are sampled uniformly. Barak et al. observed that the loss curve for neural networks trained on this task exhibits a sharp drop after an initial plateau, a case of the sort of "emergence" which has been observed in LLMs [28, 29] and in other toy settings [30].

Michaud et al. [19] extended the task to *multitask sparse parity* (MSP). In the MSP problem, input strings consist of $m + n$ bits, where the first $m$ bits are called "control bits" and the last $n$ bits are called "task bits". These leading $m$ control bits encode which "subtask" must be solved in each problem instance. Instead of a single length-$k$ set of indices $I$, a collection of $m$ sets of task bit indices (typically of equal length $k$) is chosen $\{I_1, \ldots, I_m\}$. For each sample, only one control bit is ON (1) while the others are OFF (0). If control bit $t$ is ON, then that sample's label is the parity of bits $I_t$: $y = \oplus_{l=1}^{k} x[(I_t)_l]$. Michaud et al. [19] found that when subtasks are power law distributed in frequency, where the probability that control bit $i$ is ON is $p_i \propto i^{-(\alpha+1)}$, the mean loss exhibits power-law scaling [31, 32] while individual subtasks are learned at different times proportional to their frequency [33]. They conjecture that LLM learning dynamics may be similar.

One of the main limitations of the MSP task, and the associated model of neural scaling from [19], is that subtasks are independent. However, the world, and the problem of learning from it, intuitively has a hierarchical structure—in order to learn to do certain concepts and tasks, we must first learn simpler concepts and tasks. Accordingly, humans are taught in a curriculum, where simpler concepts and tasks are learned first, and then composed into more complex ones later. To capture this structure, we now introduce the toy task *compositional multitask sparse parity* (CMSP). We show some CMSP samples to the right:

**CMSP Samples: m=3, n=9, k=3**

| | $x$ | $y$ |
|---|---|---|
| Atomic | 100101001010 | 0 |
| | 100010100011 | 1 |
| | 010110111110 | 1 |
| | 001101101110 | 0 |
| Compositional | 110001101011 | 1 |
| | 110111001010 | 0 |
| | 011110001110 | 1 |
| | 111000110100 | 1 |

Control Bits    Task Bits

CMSP is similar to MSP, except that (1) we require subtask indices to be disjoint: $I_i \cap I_j = \emptyset$ if $i \neq j$, and (2) multiple control bits can now be ON at the same time, in which case the label is the parity of the bits in the union of the indices for each subtask. If control bits $i, j$ are both ON, the label is the parity of the task bits at indices $I_i \cup I_j$. We call samples for which only one control bit is ON "atomic" and samples for which multiple control bits are ON "composite". So if $k = 3$ for all subsets $I_i$, then on atomic samples networks must compute the parity of 3 input bits, on composite samples with two control bits ON the label is the parity of 6 input bits, and so on. Above, we illustrated some CMSP data samples with $m = 3$, $n = 9$, and $k = 3$.

## 3.2 Learning dynamics on CMSP

We find that neural network training on CMSP exhibits extremely strong curriculum learning effects. To denote different types of CMSP samples, we list the ON control bits for those samples. So, given a choice of $m, n, k$ and $I_1, \ldots, I_m$, $\mathcal{D}_{\{0\}}$ denotes all samples for atomic subtask 0, and $\mathcal{D}_{\{0,1,2,3\}}$ denotes all samples from the composite subtask where the first four control bits are ON. For each subtask, there are $2^n$ possible samples in that subtask, since the $m$ control bits are fixed but the $n$ task bits are free.

We first train ReLU MLPs with 1-2 hidden layers of width 128 with the Adam optimizer on CMSP samples with $m = 4$, $n = 64$, $k = 4$, and 2000 samples per task (atomic or composite) per batch. We show the learning dynamics of these networks in Figure 2. We find that when we train on a dataset containing both atomic and composite samples $\mathcal{D}_{\{0\}} \cup \mathcal{D}_{\{1\}} \cup \mathcal{D}_{\{2\}} \cup \mathcal{D}_{\{3\}} \cup \mathcal{D}_{\{0,1,2,3\}}$ in equal proportion, atomic tasks are learned before composite ones.

Something interesting happens when we remove atomic samples from the dataset: we find that learning composite tasks takes dramatically longer. When we train on $\mathcal{D}_{\{0\}} \cup \mathcal{D}_{\{1\}} \cup \mathcal{D}_{\{2\}} \cup \mathcal{D}_{\{3\}} \cup \mathcal{D}_{\{0,1,2,3\}}$ with 10000 samples per batch (2000 per subtask), across 40 seeds, we find that 27/40 networks converge on composite subtask $\mathcal{D}_{\{0,1,2,3\}}$ within $2 \times 10^9$ samples, and the networks that

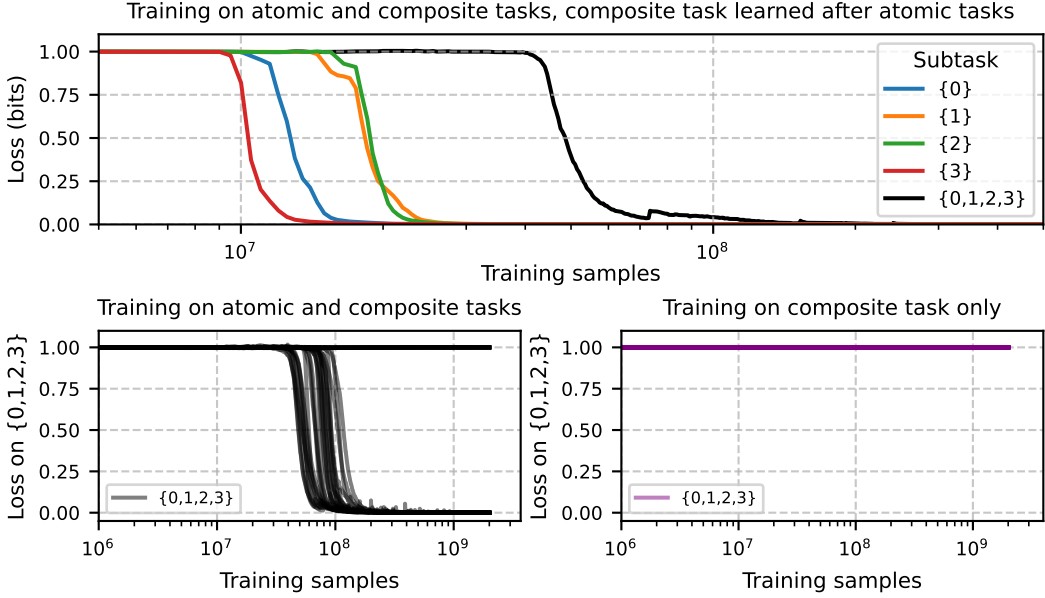

Figure 2: Training dynamics on compositional multitask sparse parity. **Top**: training dynamics for a single network trained on four atomic subtasks {0}, {1}, {2}, {3}, and their composition {0,1,2,3}. **Bottom left**: loss on compositional subtask {0,1,2,3}, when training on atomic subtasks and their composition, as in the top subplot, across 40 network seeds (depth 3, width 128). **Bottom right**: loss on compositional subtask {0,1,2,3}, when only training on samples from that composite task, *without* also training on atomic tasks, across 40 network seeds (depth 3, width 128). We find that removing the atomic tasks prevents our networks from learning the composite task. We report the minimum loss within the previous 100 steps of training to filter out loss spikes.

do converge typically converge within $2 \times 10^8$ samples (bottom left). However, when we train just on $\mathcal{D}_{\{0,1,2,3\}}$, with a batch size of 2000 samples, we find that 0/40 networks converge within $2 \times 10^9$ samples in Figure 2 (bottom right). It is much more efficient to train on a broader distribution in order to learn subtask $\mathcal{D}_{\{0,1,2,3\}}$ than just training on $\mathcal{D}_{\{0,1,2,3\}}$ on its own.

This result makes sense given the exponential hardness of learning parities [34]. Since the atomic task bits are disjoint in CMSP, networks can compute the parity of the composite tasks by computing the parity of features learned for the atomic tasks, reducing the complexity of the problem. For instance, one could imagine that depth-3 networks would compute the parity of the atomic tasks in the first layer, then the composite tasks in the next layer. Curiously though, we find that single-hidden-layer (depth-2) networks can still take advantage of a curriculum in Appendix A. We leave a mechanistic analysis of how networks learn these tasks to future work.

These sorts of dynamics may be a part of the explanation for why large-scale general-purpose models perform so strongly at many narrow, valuable tasks. We emphasize, however, that the toy task where we observe these curriculum effects is fairly contrived. It is not clear to what extent there are similarly strong effects on real-world tasks, and in many domains it is in fact possible to train narrow specialized models by only training on data in that domain. For instance, self-driving cars do not need to be trained on a broad corpus of text like LLMs.

Our work shows that for *some* types of tasks, it may be necessary to train on a broad data distribution in order to learn some subtasks efficiently. Accordingly, we now turn to the question of how to efficiently transfer knowledge from large models into smaller specialized ones.

### 3.3 Nonlocal representations in CMSP networks

If the circuitry for some subtasks were localized to a particular set of neurons, and the circuitry for other subtasks were localized to different neurons, then the task of specializing broad networks into narrow ones would be trivial. One could simply prune away neurons (or other model components, e.g. attention heads in transformers) associated with some subtasks, and keep others. However, often the situation seems more complicated than this ideal, which will be a focus for the rest of this paper.

A related problem has recently been studied in neural network interpretability, where it has been observed that individual computational units in neural networks, such as neurons, are *polysemantic*, activating across a wide variety of unrelated inputs [35, 36]. Accordingly, many assume that the true model "features" do not align with architectural components like neurons. Multiple explanations have been proposed for this phenomenon. One is simply that the model architecture does not always "privilege" a particular basis [37, 38], though other incidental reasons for polysemanticity have also been proposed [39]. Another explanation of polysemanticity is the *superposition hypothesis* [40, 36]. The superposition hypothesis suggests that the need to represent more features than there are dimensions or neurons prevents features from being represented as orthogonal directions in the feature space, and therefore all features cannot be aligned with standalone model dimensions or neurons. Recently, studies that use sparse autoencoders to identify monosemantic model features have found that most features are highly distributed across a large number of dimensions of activation space [16].

In our CMSP networks, we observe a related problem. By default, without any explicit regularization, there is no incentive for the network to localize certain circuits cleanly into a particular set of neurons. We train networks with two hidden layers on a dataset of CMSP samples with $m = 6$, $n = 18$, $k = 3$ with two different skill trees: {0}, {1}, {2}, {0,1,2}, and {3}, {4}, {5}, {3,4,5} until convergence. In Figure 3, we visualize the connectivity of a 2-hidden-layer MLP trained on this dataset, and see that the network is densely connected without obvious structure.

We attempt to prune this network to retain performance on subtask {0,1,2} while unlearning {3,4,5}. With MLPs, our groups of parameters $g$ are the in-weights, bias, and out-weights for each hidden layer neuron, so that $|G|$ is the total number of hidden neurons. We compute ablation scores on the distribution $\mathcal{D}_N = \mathcal{D}_{\{0,1,2\}}$ (estimated on 2000 samples) and prune greedily, as described in Section 2. In Figure 3, we show accuracy on subtask {0,1,2} vs. sparsity with this pruning strategy. When applied naively, network performance tends to degrade quickly, since without explicit regularization the network is not optimized to be naively prunable (see in Appendix Figure 8 and Figure 9 for more on how ablation scores for each subtask vary across neurons). However, when we perform an additional 1000 steps of training on $\mathcal{D}_{\{0,1,2\}}$ (5000 samples per batch) after pruning (after either removing neurons from the architecture or pinning pruned weights at zero) we can recover

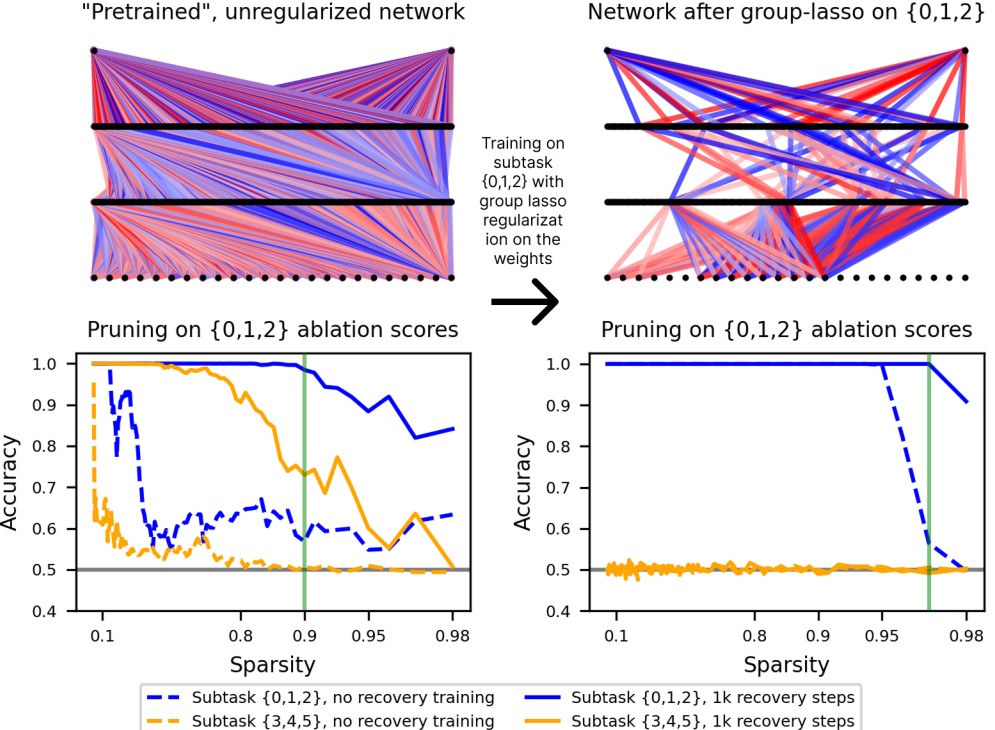

Figure 3: **Top**: We visualize the connectivity of 2-hidden-layer MLPs trained on a CMSP distribution with subtasks {0}, {1}, {2}, {0,1,2}, {3}, {4}, {5}, {3,4,5}, visualized before (**left**) and after (**right**) regularizing network weights with the group lasso sparsity penalty while training on subtask {0,1,2}. We find that network connectivity becomes sparse after regularizing. Negative weights are shown in blue and positive ones in red, with width proportional to the norm of the weight. **Bottom**: we show how pruning affects task performance on subtasks {0,1,2} and {3,4,5} at varying sparsity levels. We prune neurons based on the absolute change that ablating them has on the loss on subtask {0,1,2}. We find that subtasks here are nonlocal and entangled in the "pretrained" network (**left**). As we prune neurons according to their relevance on subtask {0,1,2}, at the sparsity at which performance on subtask {0,1,2} accuracy drops below 98% (green line), we can still recover some performance on subtask {3,4,5} with a small amount of additional training. Thus naive pruning here has not completely and robustly unlearned subtask {3,4,5}. However, after regularizing the weights (**right**), we find that not only we can more aggressively prune the network, but we have also robustly unlearned subtask {3,4,5}. Note that the degree to which subtasks are nonlocal and entangled in the pretrained networks depends on seed and width, and we show a variety of additional curves in Figure 10.

performance at higher sparsity levels. Since we are training just on composite samples, 1000 steps is not enough to re-learn this task on its own, so if we can recover performance, that will be because the mechanisms for the task were somewhat preserved after pruning. We observe that often, the tasks {0,1,2} and {3,4,5} are *entangled* – when we prune as aggressively as we can while being able to recover performance on subtask {0,1,2}, we can often still recover some performance on subtask {3,4,5}. However, for our CMSP networks the degree of entanglement is highly seed-dependent, and sometimes the subtasks are disentangled enough that pruning as aggressively as possible on one subtask does in fact robustly unlearn the other. We show pruning curves across seeds and widths in Appendix Figure 10.

### 3.4 Regularizing to "narrow" networks

We find that we can use a simple regularization penalty to simultaneously unlearn some tasks while incentivizing the network to move its features to be less distributed, allowing for more aggressive pruning. As described in Section 2, we simply perform additional training on $\mathcal{D}_{\{0,1,2\}}$ with a group

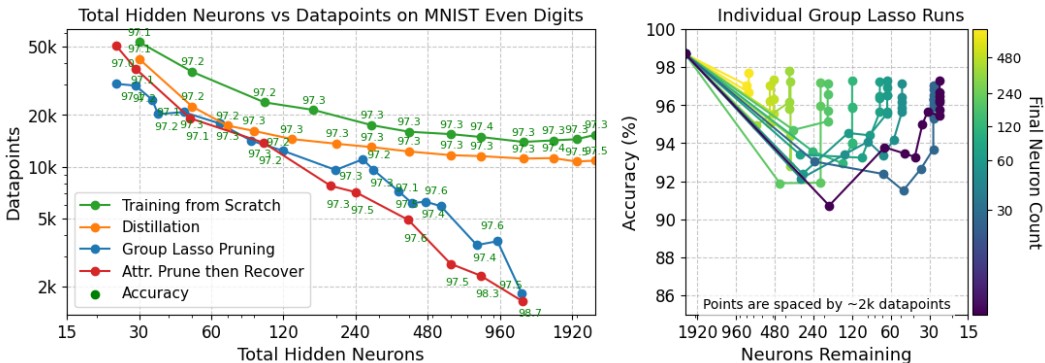

Figure 4: **Left**: We compare the performance of distillation, training from scratch, and two pruning approaches for creating small networks that classify MNIST even digits. Pruning-based approaches Pareto-dominate distillation, achieving high compression ratios with fewer datapoints. All points are averaged over 10 individual training runs. **Right**: When pruning using group lasso, it often helps to first prune rapidly, degrading performance, and then recover performance with no regularization. Each line represents a single training run, with a new point logged every 2,000 datapoints. Darker lines correspond to more aggressive pruning runs that attain lower final neuron counts.

lasso sparsity penalty on the network weights. We aim for this penalty to "clean up" circuitry [41] not relevant to prediction on $\mathcal{D}_{\{0,1,2\}}$ while also sparsifying the weights across groups of parameters, allowing for easier pruning.

We apply this regularization to our CMSP networks, training with penalty strength $\lambda = 10^{-3}$ with a batch size of 2000 for 10000 steps. With this regularized network, we then re-compute neuron ablation scores and prune. In Figure 3 we find that, in CMSP networks, this method is effective at unlearning skills and making the network more prunable. We find that we can prune more aggressively while retaining performance on subtask {0,1,2}, and we are not able to recover performance on {3,4,5} at any sparsity level. These results complete our case study on CMSP: in order to efficiently learn compositional tasks, we train on a broad distribution of tasks. Then, we can regularize the network to turn this broad network into a much smaller one which nevertheless retains the broad network's performance on the target compositional task.

## 4 Pruning vs distillation: MNIST

We now consider the problem of creating narrow systems in more natural domains, first on MNIST. As our narrow subtask, we choose only even digits from the original MNIST dataset [42]. We compare the resources required to achieve good performance on this narrow task when (1) training from scratch on the narrow task, and (2) distilling models from a general teacher on this task, (3) using group-lasso regularization to prune a large general model and (4) using attribution-based pruning and then recovery training on a large general model. Our teacher model is a ReLU MLP with two hidden layers each of width 1200, as in [26], and achieves 98.7% accuracy on the test set. When pruning, we use this same teacher model as our initial model and prune its hidden neurons to create a smaller network.

When using distillation (2), we use the approach of Hinton et al. [26] with $T = 20$. When pruning with the group lasso penalty (3), we use $\lambda$ values ranging from 0.001 to 0.008. Unlike in Section 3.4, we regularize and prune simultaneously, pruning neurons when their L2 norm drops below 0.05. When the number of remaining neurons drops below a target threshold, we remove the pruning penalty and continue training to recover lost performance during pruning. When we prune up front and then separately recover performance (4), we use attribution scores as described in Section 2.

To compare methods, we require that each method reach a test-set accuracy of 97 percent, and we then plot the frontier of neuron count versus datapoints subject to that threshold. As seen in Figure 4 (right), it is often optimal to first aggressively prune the network down to the desired size and further train it until it reaches the requisite accuracy.

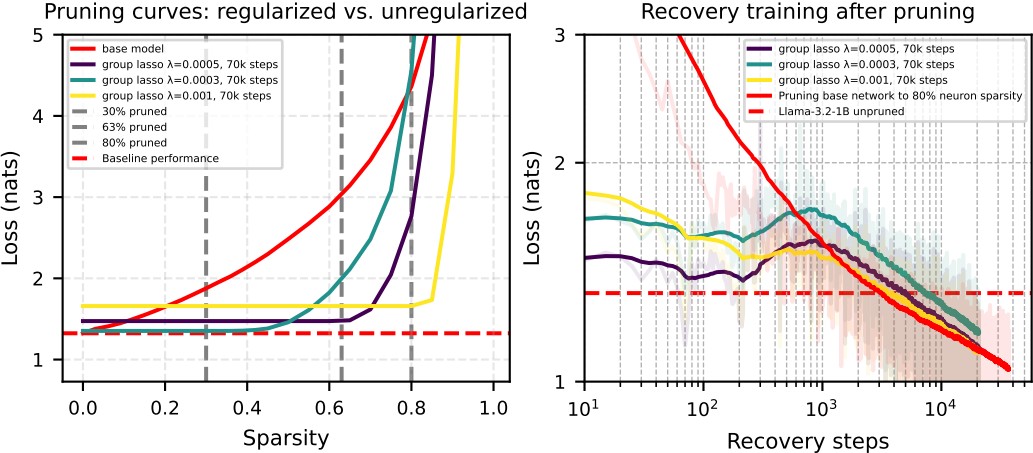

Figure 5: **Left**: Neuron sparsity vs. loss curve for LLMs finetuned with group lasso regularization with varying $\lambda$ for 70k steps, as compared to the base network. Regularization flattens the sparsity vs. loss curve, at the cost of slightly degrading model performance. **Right**: After pruning our networks to 30%, 63%, and 80% sparsity for our runs with $\lambda$ of 5e-4, 3e-4, and 1e-3, respectively, we recover performance with additional training. We find that we can recover performance lost during pruning, including in the network that was pruned without first using group-lasso regularization training. The plot shows an exponential moving average of batch losses, with individual batch losses shaded in back.

We find that while group lasso pruning is highly sensitive to choices in hyperparameters, both pruning methods Pareto-dominate distillation and training from scratch, especially at high neuron counts. Moreover, while other methods cannot consistently bridge the 97 percent accuracy threshold with fewer than 25 hidden neurons, aggressive pruning can consistently shrink the network's size to a lower absolute limit.

## 5    Pruning vs distillation: LLMs on Python documents

We next study LLMs. As our narrow task $\mathcal{D}_N$, we choose next-token prediction on Python documents in the GitHub Code Dataset.

We first prune the neurons in the MLP blocks of Llama-3.2-1B [43]. Later, we will also consider pruning residual stream dimensions, which involves pruning all model parameters that "read to" or "write from" a dimension of the residual stream [37]. We use attribution scores when pruning, and show that the attribution scores correlate moderately with true ablation scores in Appendix Figure 11. In Figure 5 (left), we show neuron sparsity vs loss curves. When pruning neurons naively, we see that loss increases quickly at low sparsity levels. We also experiment with applying group lasso regularization while further training on Python documents (learning rate of 2e-6, max length 512, 18 documents per batch) and find that this training does indeed level out the sparsity vs. loss curve, albeit at a slight cost to the loss. Fortunately, we find that we can recover lost performance after pruning by doing a small amount of additional training on Python documents in Figure 5 (right). We find that despite the loss increasing substantially after naively pruning, we can also quickly recover that lost performance, and overall this strategy seems to be more efficient than using group lasso training. We therefore next compare naive pruning + recovery training against distillation and training networks from scratch.

We train networks with the Llama 3 [43] architecture of varying shape and size. We use a learning rate of 5e-4, sequence length of 1024, and batch size of 64. For distillation, we use Llama-3.1-8B as a teacher with $T = 2$. For pruning + recovery, we prune Llama-3.2-1B to varying levels of neuron and residual stream sparsity, shown in Appendix Table 3. In Appendix Figure 12 we show learning curves for these three approaches. In Figure 6, we tentatively find that pruning substantially outperforms training from scratch and distilling a model from scratch on the data-parameter frontier. Given a

target narrow network size and a fixed data budget, if one already has access to a general model, it appears to be more efficient to prune that model than it is to perform distillation.

We also experiment with pruning random model components instead of components selected by attribution score and find that random pruning performs similarly [44]! In Appendix Figure 13, we find that after a moderate number of recovery steps, attribution pruning and random pruning result in the same recovered performance. This result is in line with our earlier discussion of nonlocality, and the empirical findings of Bricken et al. [16], that monosemantic features are distributed widely across model components. While some studies have found some geometric similarity between functionally similar features [45, 46], in our case it does not seem like the relevant features for our task are localized into a set of "Python" vs. "non-Python" neurons, at least that attribution pruning identifies.

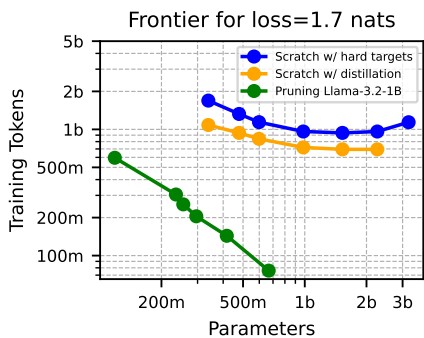

Figure 6: Frontier of network parameters vs. data required to achieve a certain cross-entropy loss on Python documents. For the task of creating a LLM specialized on Python documents, we find that pruning Llama-3.2-1B and then performing recovery training is much more efficient than training LLMs from scratch or distilling LLMs from scratch on the soft targets of Llama-3.2-8B.

**Unlearning**: We lastly evaluate whether neuron pruning can robustly unlearn downstream tasks. We consider three pruning methods applied to Llama-3.2-1B at sparsity levels of 30%, 63%, and 80%: (i) random neuron pruning, (ii) attribution-based pruning on Python code, and (iii) models trained with group-lasso regularization on Python documents from Figure 5. We evaluate each pruned model on four benchmarks: counterfact [47], AI2-ARC (ARC-Easy) [48], WMDP-Bio, and WMDP-Cyber [49] for hazardous knowledge. We found that the base model performed poorly on WMDP-Chem. For each model, we first measure baseline accuracy after pruning, then fine-tune on each benchmark's training set with batch size 4, sweeping learning rates between 1e-6 and 1e-4 and reporting the best-performing configuration on the test set. In Table 1, we find that all pruning techniques, including random pruning, robustly unlearn non-Python skills with roughly similar effectiveness, except that at low sparsity, attribution pruning does not achieve much unlearning. This result contrasts somewhat with our findings on CMSP networks in Sections 3.3 and 3.4, where training with the group lasso penalty led to stronger unlearning than naive pruning.

Table 1: **Unlearning**: We prune Llama-3.2-1B using various techniques to create narrow Python-specific models, then evaluate on other tasks to assess whether pruning led to unlearning. Accuracies are reported for the pruned model before finetuning and after finetuning on a train set of each benchmark (non-FT→FT). ✓ indicates successful unlearning (FT score ≥10% lower than base).

| Method | Spar. | counterfact | AI2-ARC | WMDP Bio | WMDP Cyber |
|---|---|---|---|---|---|
| Base | 0% | 0.18→**0.49** | 0.65→**0.67** | 0.52→**0.62** | 0.35→**0.59** |
| Random | 30% | 0.00→**0.50** | 0.24→**0.25**✓ | 0.22→**0.27**✓ | 0.27→**0.27**✓ |
| Attribution | 30% | 0.12→**0.65** | 0.35→**0.69** | 0.36→**0.56** | 0.27→**0.55** |
| Group Lasso | 30% | 0.06→**0.47** | 0.25→**0.25**✓ | 0.24→**0.31**✓ | 0.26→**0.27**✓ |
| Random | 63% | 0.00→**0.45** | 0.25→**0.25**✓ | 0.23→**0.27**✓ | 0.26→**0.28**✓ |
| Attribution | 63% | 0.02→**0.52** | 0.25→**0.30**✓ | 0.23→**0.34**✓ | 0.26→**0.32**✓ |
| Group Lasso | 63% | 0.03→**0.45** | 0.25→**0.25**✓ | 0.24→**0.29**✓ | 0.27→**0.30**✓ |
| Random | 80% | 0.00→**0.42** | 0.26→**0.25**✓ | 0.26→**0.28**✓ | 0.23→**0.26**✓ |
| Attribution | 80% | 0.00→**0.47** | 0.25→**0.25**✓ | 0.23→**0.27**✓ | 0.26→**0.28**✓ |
| Group Lasso | 80% | 0.02→**0.44** | 0.25→**0.25**✓ | 0.24→**0.29**✓ | 0.26→**0.29**✓ |

## 6 Related Work

**Distillation.** Many works have built on the original distillation work of Hinton et al. [26], seeking to transfer intermediate representations [50, 51] and applying these techniques to language models [52, 53], often after pruning [54, 55]. Relevant to our discussion, Turc et al. [56] showed that pretraining the student before distillation can substantially improve results.

**Pruning.** Even early approaches to pruning used second-order methods for pruning weights [57, 58], whereas our "attribution" scores are first-order. When training vision models, Zhou et al. [25] and Wen et al. [24] used a group lasso penalty like we do, albeit while training on the full data distribution. Sanh et al. [59] proposed "movement pruning" to prune weights during transfer learning. A variety of works have applied structured pruning to LLMs [60, 61], including Xia et al. [62] who develop a method for task-specific pruning. Vural and Erdogdu [63] provide evidence that pruning boosts sample efficiency by directing gradient updates to a sparse set of important directions.

**Task Structure and Learning Dynamics.** Several works have lately investigated the relationship between task structure and learning dynamics [12–15]. Liu et al. [64] also briefly study a task similar to CMSP to show that hierarchical relationships between tasks cause what they call "domino" learning dynamics. We also highlight the "Skill it!" paper of Chen et al. [65], who empirically investigate hierarchical learning dependencies for real LLM tasks.

**Machine Unlearning.** Many approaches to unlearning have been proposed [66–70]. Guo et al. [71] study how applying unlearning fine-tuning to different model components affects unlearning success, inspired by a mechanistic understanding of how knowledge is retrieved in LLMs. Lee et al. [72] show that distillation can be applied after unlearning fine-tuning to more robustly remove unwanted capabilities. Highly relevant is the work of Cloud et al. [73], who apply "gradient routing" during training to localize capabilities to different model components, allowing pruning to be used for unlearning.

# 7    Discussion

**Limitations**: This work explored some aspects of neural network learning which are relevant to the creation of narrow AI systems. Much work remains to be done in determining how relevant our findings are to real-world networks in various domains. We first emphasize that CMSP is a toy task, an existence proof that curriculum learning effects can be quite strong, but it is still unclear whether these dynamics occur on more natural datasets. For our pruning experiments, we emphasize that our greedy attribution pruning approach is quite simple, and we cannot rule out that more sophisticated pruning strategies would be more successful in preserving some skills while unlearning others [74]. Also, we did not optimally scale hyperparameters in our LLM experiments, so the performance of each of our models in Section 5 is likely imperfect. While we found that pruning a pre-existing Llama model was more efficient than training a model from scratch on Python documents, this could be due to our imperfect training setup, and not because one needs to train on a broad distribution of data to learn Python documents due to curriculum effects.

**Conclusion**: In this work, we have studied some potential challenges involved in creating narrow AI systems, having to do both with the structure of data and the structures learned internally by neural networks. Underlying this work is a perspective from *mechanistic interpretability*, that neural networks compute a variety of sparse *features* [35], each with a distributed representation [36, 16, 46], and that these features are computed from each other hierarchically in *circuits* [75–77]. First, we found that in order to learn certain complex features, we may have to first train on a broad set of samples which encourage the learning of simpler features. Second, because features are distributed across model components, it is a nontrivial problem to move a set of task-specific features and circuits into a smaller network, especially while *unlearning* unneeded circuits.

While a neural network's computation across the whole data distribution may be quite complex, we hope that the computation that networks perform on any particular task will be reducible to something less complex. That less complex computation, whatever it is, might be interpretable as a circuit [76, 77], or reduce to a simple program [78], or, as we studied here, could be instantiated in a much smaller network. However, this hope, and the question of whether it is possible to create narrow-purpose versions of today's models, probes at a basic question about the nature of the intelligence of these models. If their apparent generality indeed results from their having learned a large, diverse set of crystallized, task-specific circuits, then we ought in principle to be able to create competent, specialized versions of these models by just transferring the relevant task-specific circuits. However, if the intelligence of our models, and intelligence more generally, is better understood as resulting from a single unified algorithm, then the basic prospect of creating narrow AI systems that are as strong as truly general ones could be a challenge.

## Acknowledgments and Disclosure of Funding

We thank Ziming Liu, Josh Engels, David D. Baek, Tom McGrath, and Jamie Simon for helpful conversations and feedback. E.J.M. was supported by the NSF via the Graduate Research Fellowship Program (Grant No. 2141064) and under Cooperative Agreement PHY-2019786 (IAIFI).

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

## A  Additional results on CMSP

Here we include some additional results on CMSP. We first explore whether network shape influences learning dynamics when learning composite tasks (w/ corresponding atomic tasks in the training dataset) vs. when learning an atomic task on its own. We find that the effect of depth is similar between composite vs. atomic tasks. This suggests that our networks do not necessarily need to be deep in order to take advantage of compositional structure in this task. Note that a previous version of this manuscript suggested the opposite–we had observed that deeper networks learn composite CMSP tasks faster than shallow networks, but it turns out that deeper networks also learn standalone atomic tasks faster than shallow networks. We show loss curves of depth-3 width-128, depth-2 width-361, and depth-2 width-128 networks in Figure 7.

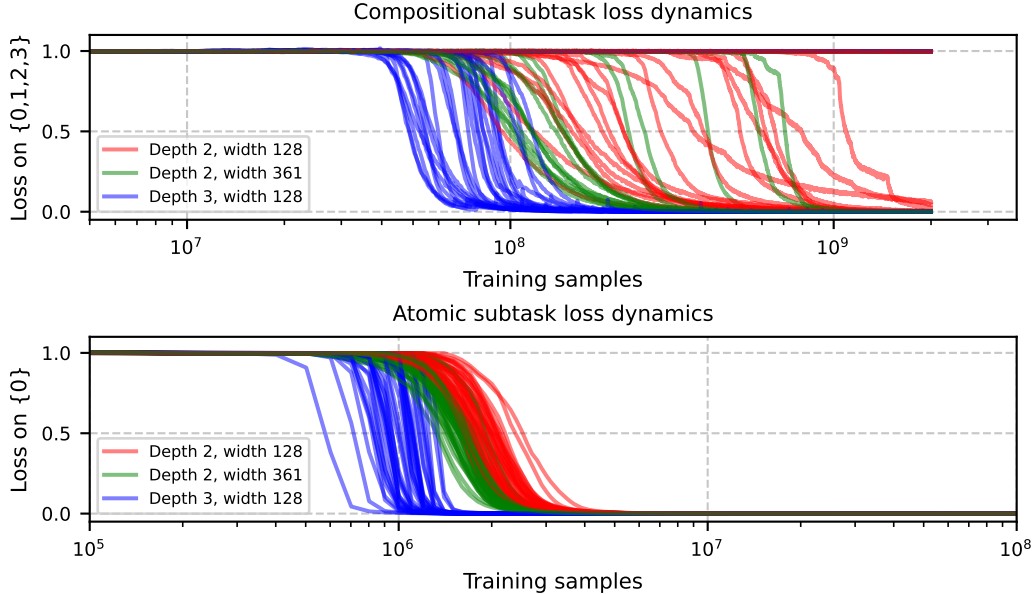

Figure 7: Impact of network shape on CMSP learning. **Top**: Forty runs of training with different seeds on CMSP dataset $\mathcal{D}_{\{0\}} \cup \mathcal{D}_{\{1\}} \cup \mathcal{D}_{\{2\}} \cup \mathcal{D}_{\{3\}} \cup \mathcal{D}_{\{0,1,2,3\}}$ with $m = 4$, $n = 64$, $k = 4$. We find that 27/40 runs converge with two hidden layers versus 19/40 runs with a single hidden layer with width 128 and 16/40 runs with a single hidden layer and width 361. Single-hidden-layer networks are therefore able to learn composite subtasks in CMSP roughly as well as deeper networks. **Bottom**: Forty runs of training on a standalone atomic task $m = 1$, $n = 64$, $k = 4$.

We also show how ablation scores vary across neurons in our unregularized networks. With the setup in Section 3.3, we show ablation scores for each neuron for each subtask in Figure 8 and Figure 9. We find that there are neurons which have high scores across most subtasks, even subtasks in different "skill trees" ({0,1,2} versus {3,4,5}).

In Figure 10 we show sparsity vs accuracy curves from pruning like in Figure 3 (bottom left), across seeds and network sizes. In our unregularized networks, there is a lot of variation between runs in how entangled the different subtasks are.

## B  Additional results on LLMs

### B.1  Additional plots

We also show some additional results with LLMs. In Figure 11, we show how attribution scores correlate with ablation scores in LLMs. We show results for pruning both neurons and residual stream dimensions.

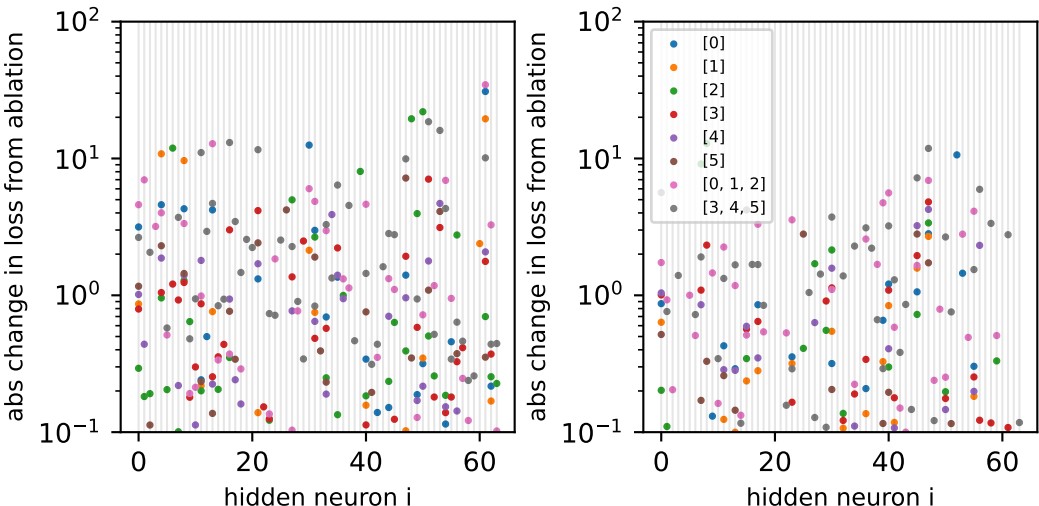

Figure 8: For each hidden neuron in the first hidden layer (**left**) and the second hidden layer (**right**) in a CMSP network, we show the ablation score of that neuron for each subtask. For some neurons, ablation scores are high across multiple subtasks, even from separate "skill trees".

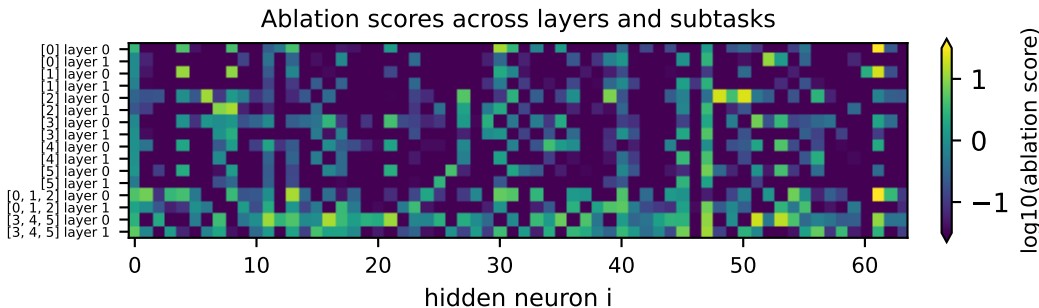

Figure 9: Ablation scores for each subtask and each neuron. This is simply a different way of visualizing the results in Figure 8.

In Figure 12, we show learning curves across runs when training from scratch on Python documents, when performing distillation, and when performing recovery training after pruning both neurons and residual stream dimensions, to varying target sparsities, of Llama-3.2-1B.

In Figure 13, we show recovery curves after pruning neurons and residual stream dimensions from Llama-3.2-1B to varying sparsity levels. We find that pruning random components performs roughly as well as pruning components with the lowest attribution scores.

### B.2 Additional experimental details

When training networks from scratch and distilling from scratch (Figure 12 and Figure 6, we train transformers with the Llama 3 architecture [43] of varying size, listed in Table 2.

When we prune Llama-3.2-1B, we prune neurons and residual stream dimensions with the sparsity combinations shown in Table 3.

## C Compute estimates

We estimate the compute used for each of our experiments.

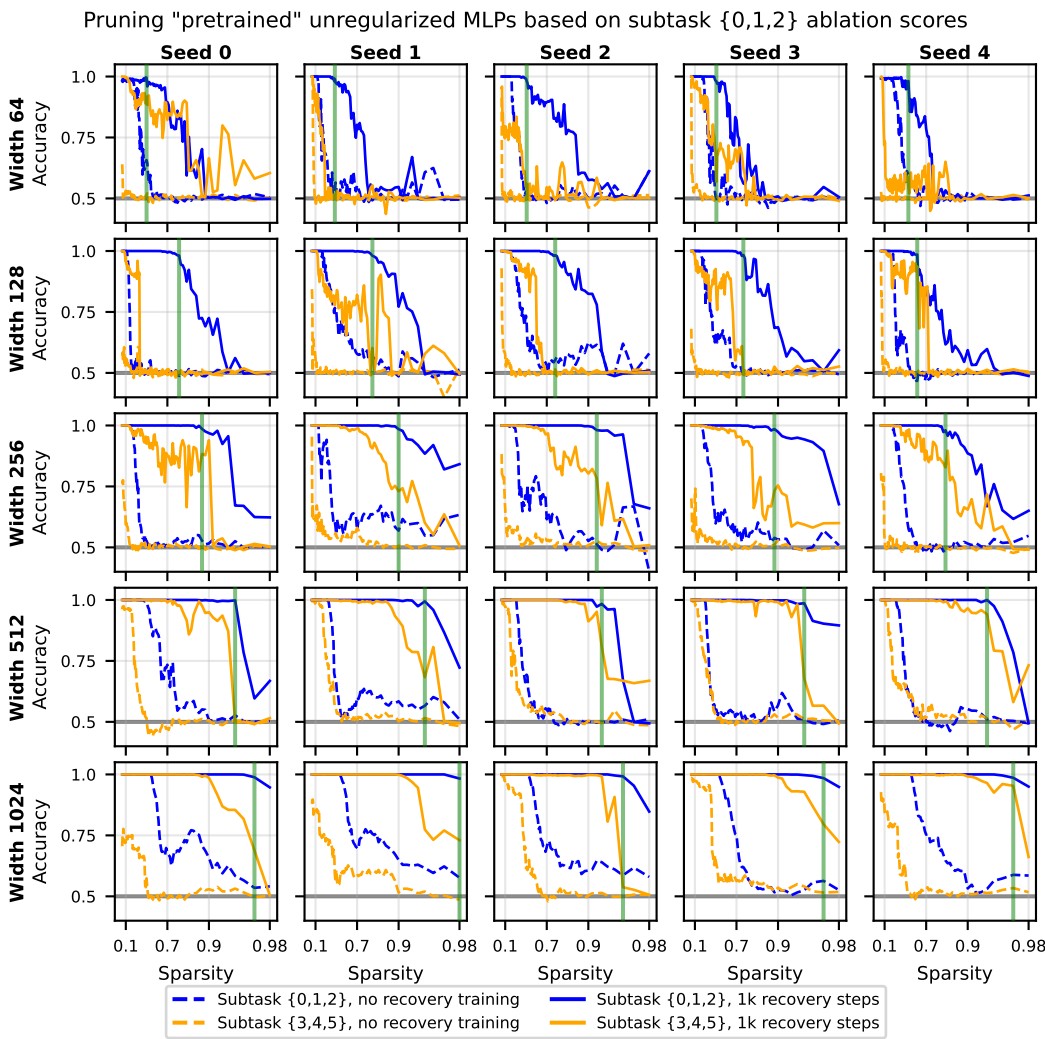

Figure 10: Pruning sparsity vs accuracy curves for pretrained CMSP networks. We train MLPs of varying width and random seed on a CMSP dataset with two skill trees, as in Figure 3. We prune neurons based on ablation score on subtask {0,1,2}. On some networks, ablating neurons to maximum sparsity while preserving performance on subtask {0,1,2} robustly unlearns subtask {3,4,5}. However, for other networks, the subtasks seem more entangled, and we can recover performance on subtask {3,4,5} even after ablating to the maximum extent we can still recover 98% accuracy on subtask {0,1,2} (green line).

For the CMSP training experiments shown in Figure 2 and Figure 7, we ran 4 configurations each with 40 different seeds. We ran each job on a GPU, but on a cluster with a variety of different node configurations and GPUs. Jobs generally took between 5-60 minutes to complete, for between 13-160 total hours of GPU time.

For the experiments shown in Figure 3 and Figure 10, each run, across 5 choices of seed and 5 choices of width, trained networks on a CMSP task, ran pruning experiments, performed group-lasso training, and pruned again. Each such run took between 5-120 minutes on our cluster with varying GPU configurations, for a total time of 2-50 hours of GPU time.

For our MNIST experiments shown in Figure 4, we plot a total of 54 points, each of which is averaged over 10 training runs. We estimate that each run took around 2 minutes on our cluster, totaling to around 18 hours of GPU time.

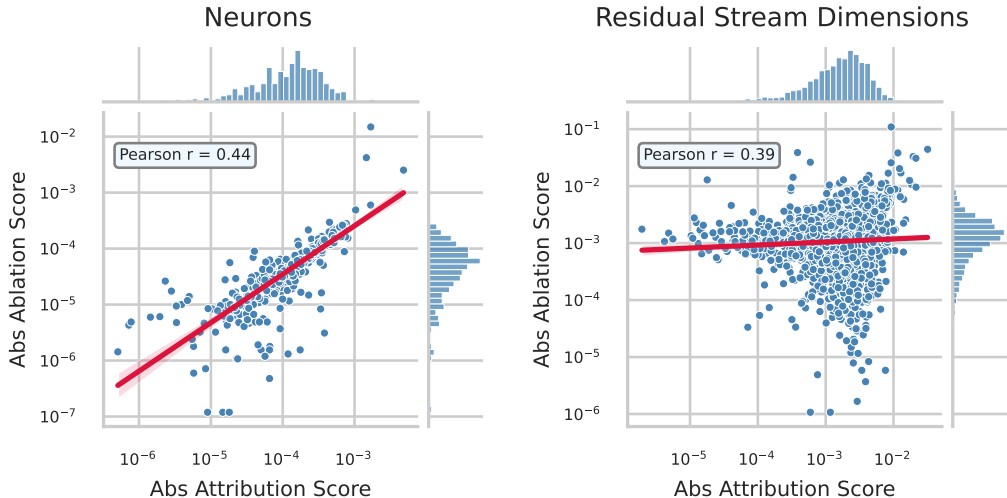

Figure 11: Comparison of ablation vs. attribution scores for Llama-3.2-1B neurons (left) and residual stream dimensions (right), evaluated on a single batch of Python code documents. For each figure, we fully ablate model components and compare the absolute change in loss (Abs Ablation Score, simply called "ablation score" elsewhere in the paper) with the absolute value of the linear estimate of this change computed from model gradients (Abs Attribution Score, called "attribution score" elsewhere). Note that the correlation between the $\log$ of these scores is 0.80 for neurons and 0.05 for residual stream components.

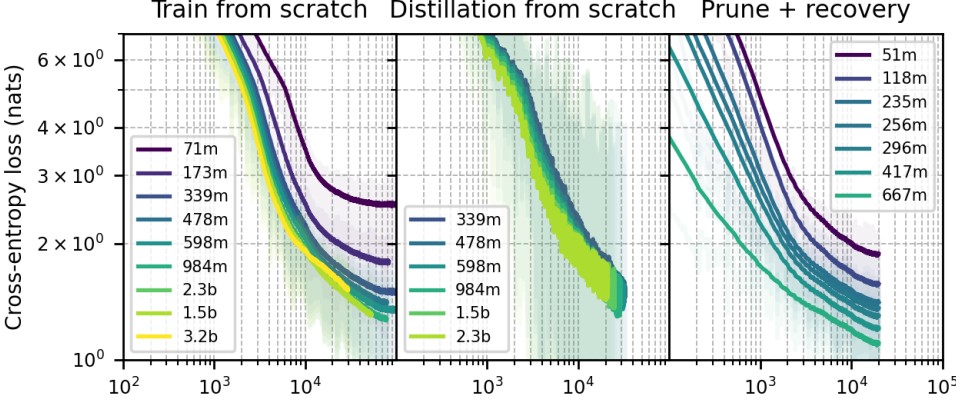

Figure 12: Training curves of LLMs on Python documents when training from scratch (**left**), training from scratch but with distillation of a larger pretrained LLM (**center**), and when training pruned pretrained models (**right**) to recover performance lost during training.

For our LLM experiments, we ran our experiments on A100-80GB nodes, with a single GPU allocated per experiment. When we trained models from scratch on Python documents, we used 9 configurations with job lengths between 1-3 days. For distillation we had 9 configurations with job lengths between 2-3 days, though some jobs failed. For the group lasso training experiments in Figure 5, we show 3 configurations trained with job lengths of 3 days, and we performed recovery training on these models with a job length of 1 day. We estimate that the total time for these jobs was less than 1600 hours of A100 time, though the full set of experiments we attempted in the work that led to this manuscript could be over 5000 hours.

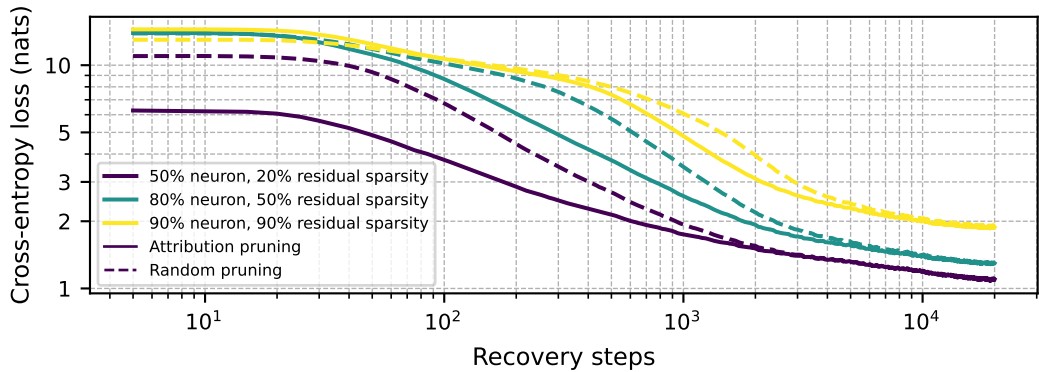

Figure 13: Recovery training curves after pruning neurons and residual stream dimensions, for Llama-3.2-1B on Python code documents. We compare recovery performance when pruning based on attribution scores vs. choosing components randomly. Attribution scores are computed across 1024 documents with a max length of 1024 tokens. We find that pruning with attribution scores is better than pruning random components, however this gap is eventually recovered. For instance, at step 5365 when our run with 50% neuron sparsity and 20% residual sparsity first matches the performance of the original model ($\approx$ 1.3 nats), the performance on randomly-pruned model is almost identical at 1.301 nats.

Table 2: Transformer model configurations explored

| Hidden size | #Layers | #Heads | Intermediate size |
|---|---|---|---|
| 256 | 4 | 4 | 1,024 |
| 512 | 8 | 8 | 2,048 |
| 768 | 12 | 12 | 3,072 |
| 864 | 16 | 16 | 3,456 |
| 1,024 | 16 | 16 | 4,096 |
| 1,280 | 20 | 20 | 5,120 |
| 1,536 | 24 | 24 | 6,144 |
| 1,792 | 28 | 28 | 7,168 |
| 2,048 | 32 | 32 | 8,192 |

Table 3: Neuron and residual sparsity configurations when using attribution pruning on Llama-3.2-1B.

| Neuron sparsity | Residual sparsity |
|---|---|
| 0.50 | 0.50 |
| 0.80 | 0.50 |
| 0.90 | 0.50 |
| 0.95 | 0.50 |
| 0.80 | 0.80 |
| 0.90 | 0.90 |

