# OpenReview forum: "On the creation of narrow AI: hierarchy and nonlocality of neural network skills"
_NeurIPS.cc/2025/Conference — NeurIPS 2025 poster_

### Official Review · Reviewer_tPS8 · 2025-06-30

**Clarity:** 3
**Significance:** 2
**Originality:** 2
**Rating:** 3
**Confidence:** 4

**Summary:**

The submission investigates the creation of narrow, specialized models. The authors argue that hierarchical tasks in some cases require broad pre-training of larger models. Experiments on synthetic data show that such cases can exist. Further, the authors evaluate whether specialized knowledge can be localized in the weights of models, and could be isolated by pruning the other weights. They find that specialized knowledge is entangled in weights, but can be dis-entangled by fine-tuning on the target data with a group-lasso regularization.

**Questions:**

See strengths and weaknesses.

**Ethical Concerns:**

["NO or VERY MINOR ethics concerns only"]

**Final Justification:**

I appreciate the author's response. It demonstrate awareness of room for improvement. To me, the paper demonstrates very good ideas, but they seem a unfinished. I encourage the authors to improve their submission, regardless of whether it get's accepted this time. For that reason, I keep my score.

**Limitations:**

yes

**Paper Formatting Concerns:**

All good.

**Quality:**

3

**Strengths And Weaknesses:**

### Strengths

+ The motivation of narrow models is convincing, especially via efficiency.

+ The paper presents an interesting case study on learning dynamics of compositional tasks. While the results are intuitive (atomic tasks learned first, composite tasks learned after), I wonder if the experiments conflates compositionality with 'hardness' of the task and can support the conclusion of hierarchical learning. Presumably, compositional tasks are significantly harder and so will be picked up later. To get clean results, the two need to be disentangled. I'm not sure if CMSP allows for that.

+ Albeit not entirely surprising, the combination of fine-tuning with l1 regularization and subsequent pruning is a reasonable idea, and the experiments support the combination. I'm not sure if similar ideas have been tried before since it seems such a natural combination, but its interesting that localization can be strenghtend post-hoc even on pre-trained models.

+ The submission is generally clear and well written. Experiments are well motivated. I especially appreciate that the authors don't embellish their results and discuss limitations.

### Weaknesses

- The motivation via mechanistic interpretability or verifiability is not convincing to me. Differences between full and narrow models are probably in scale, not in general composition. Particularly in cases like coding task, where models are qualitatively similar. Hence, methods that work on narrow models can also be expected to work on general models.

- The authors make a strong assumption that (some) skills are hierarchical. This is not my field, but I keep wondering. Hierarchical skills exist - but is there actual evidence for it? Is that relevant in practice, or is this a notion carried over from what and how humans learn? Even if it exists, its unclear to what extend the CMSP experiments represent real-world compositionally and how far findings might carry over. The authors touch upon that limitation in line 159.

- While I like Figure 5, I feel it only shows part of a picture. Specifically if random pruning achieves similar performance, it appears that the overall computational load might be lowest to just randomly prune to the desired sparsity and follow up with recovery steps. The group-lasso steps appear to keep the loss low for a low recovery step regime, but afterwards at best match the baseline.

- I keep wondering how the two parts of the paper are connected. The first considers hierarchical skills and how easy / hard they are to learn. The second studies extracting skills from a larger model into a smaller one. The two appear disjunct, albeit they may both be part of a larger pipeline. If the argument the authors wish to make is that one should pre-train broadly to learn complex composite skills faster, and then prune aggressively, I would encourage them to tie the two together more tightly in an overarching narrative. In the second half, the evaluation uses a Llama on code. Maybe running experiments on hierarchical skills in that domain might help to form a coherent narrative?

---

> ### Author Rebuttal · Authors · 2025-07-31
>
> Thank you for your thoughtful comments! We’re glad you found the paper’s motivation convincing, interesting, and that it was clear and well written. We note that in addition to group lasso regularization, we also investigate a number of other methods for creating narrow networks, such as distillation, attribution pruning, random pruning, and training from scratch as a baseline. We’ll respond to your comments below:
>
> > The motivation via mechanistic interpretability or verifiability is not convincing to me. Differences between full and narrow models are probably in scale, not in general composition. Particularly in cases like coding task, where models are qualitatively similar. Hence, methods that work on narrow models can also be expected to work on general models.
>
> Our hope is just that more narrow models will have fewer mechanisms for us to enumerate and explain! It could be worth including some additional discussion of this point at the end of the paper, along with our more philosophical point about how to understand the computation that models are doing in the first place.
>
> > The authors make a strong assumption that (some) skills are hierarchical. This is not my field, but I keep wondering. Hierarchical skills exist - but is there actual evidence for it? Is that relevant in practice, or is this a notion carried over from what and how humans learn? Even if it exists, its unclear to what extend the CMSP experiments represent real-world compositionally and how far findings might carry over. The authors touch upon that limitation in line 159.
>
> Yes it is unclear right now to what degree curriculum effects like those we observed on CMSP influence LLM learning, for instance. But we think the CMSP task is an interesting existence proof that certain datasets can have extremely strong effects like this!
>
> > While I like Figure 5, I feel it only shows part of a picture. Specifically if random pruning achieves similar performance, it appears that the overall computational load might be lowest to just randomly prune to the desired sparsity and follow up with recovery steps. The group-lasso steps appear to keep the loss low for a low recovery step regime, but afterwards at best match the baseline.
>
> Yes this is accurate! We could have made this point more clearly, but we did mention that: “We find that despite the loss increasing substantially after naively pruning, we can also quickly recover that lost performance, and overall this strategy seems to be more efficient than using group lasso training. We therefore next compare naive pruning + recovery training against distillation and training networks from scratch”. It’s possible that group lasso regularization would more effectively unlearn LLM skills vs. random pruning, but we haven’t yet tested this (but could in the discussion period next week, if you’d find that helpful!).
>
> > I keep wondering how the two parts of the paper are connected. The first considers hierarchical skills and how easy / hard they are to learn. The second studies extracting skills from a larger model into a smaller one. The two appear disjunct, albeit they may both be part of a larger pipeline. If the argument the authors wish to make is that one should pre-train broadly to learn complex composite skills faster, and then prune aggressively, I would encourage them to tie the two together more tightly in an overarching narrative. In the second half, the evaluation uses a Llama on code. Maybe running experiments on hierarchical skills in that domain might help to form a coherent narrative?
>
> Another idea we had of tying the two parts together would be to evaluate whether group lasso sparsity training leads to more effective unlearning in LLMs, like we showed in CMSP. For instance, we could fine-tune our Python-specific LLMs on a variety of other tasks and compare the networks pruned with group lasso training to vs. more naive pruning baselines. We could also run these experiments in the discussion phase if they would be of interest?

---

> > ### Comment · Reviewer_tPS8 · 2025-08-03
> > **Reviewer Response**
> >
> > I would like to thank the authors for their response. It appears they are aware of how to continue refining their submission, and I encourage them to do so, especially in testing their assumptions and fitting the two parts of the paper together. For now, I will keep my score.

---

### Official Review · Reviewer_gjJ8 · 2025-07-01

**Clarity:** 3
**Significance:** 3
**Originality:** 3
**Rating:** 5
**Confidence:** 4

**Summary:**

This paper explores the problem of building narrow AI systems, models specialized for a single task or domain, by investigating two core challenges: 1) when can a narrow model be trained from scratch 2) how do you transfer specific skills from a pretrained general model.

For challenge 1, the authors introduce a synthetic benchmark task, compositional multitask sparse parity (CMSP), designed to exhibit hierarchical dependencies among subtasks. Using CMSP, they show that training on broader data distributions can enable more efficient learning of composite tasks.

For challenge 2 the paper compares methods for narrowing models with a focus on pruning, distillation and group lasso regularisation - a regularisation method introduced by the authors to encourage the model to learn tasks in a way that better supports pruning. The authors report that pruning + recovery training (continued training on the desired task) often outperforms distillation and training from scratch at preserving desired skills while reducing model capacity. Experiments are conducted on CMSP, MNIST and Python documentation using LLMs.

**Questions:**

- Have you done any exploration into tasks that have a hierarchical structure in the presentation of the data but not necessarily the task? For example the use of abstractions in image based tasks, where there is an underlying structure that can make the task easier but is not directly tied to the result.
- By removing the ability to recover performance on a different task, is it possible to degrade performance on your current task if they are closely linked? This seems to me to be a bit of a random proxy to be optimizing for that could possibly lead to degradation of the whole system if done in some settings.
- As the paper stands, I would raise my score if the figures are better labelled and explained, the language and discussion was made a bit more rigorous and if the limitations section was a more critical reflection of the work actually done in this paper.

**Ethical Concerns:**

["NO or VERY MINOR ethics concerns only"]

**Final Justification:**

I think this is an interesting piece of work as I said in my previous review. Given the commitment to improving the presentation I am happy to raise my score.

**Limitations:**

The authors did not efficiently evaluate the limitations of their work. They show the "simple pruning strategy" can outperform distillation so claiming this as a limitation is a bit hollow. Similarly, the LLM is used as a baseline so while the results may change if the LLM was well tuned the lessons learnt from the LLM experiment should still hold true. The rest of the limitations is just philosophical musings that are more a statement about the subfield of hierarchical learning and not about the work. Instead the authors could have mentioned their lack of task diversity in their synthetic study, lack of evaluation of group lasso regularisation in the LLM experiment and the fact that much of their analysis is done only on synthetic tasks.

**Quality:**

2

**Strengths And Weaknesses:**

**Strengths**
- The paper focuses on narrow AI systems, something that is less common after the recent successes of general-purpose models. This is still an important area of research as we have evidence from biological agents of the benefit of the combination of smaller, focused models and so definitely deserves further exploration.
- The CMSP benchmark nicely introduces a hierarchical dependency between tasks and can act as a useful tool for future investigations of tasks with this very hierarchical structure.
- The paper focuses on an interesting problem that is often pushed aside in current works, the overlapping neuron representations of different skills in larger models, which can make pruning challenging and reduces explainability.
- The paper is laid out well, building both the narrative of their method and the use of their experiments to help the reader understand both the challenges of this problem and the proposed solutions.

**Weaknesses**
- The paper claims to explore when it is possible to train narrow models from scratch and concludes that "Our work shows that for some types of tasks, it may be necessary to train on a broad data distribution in order to learn." The authors only investigated one type of task structure - highly hierarchical - and found you needed a broad dataset to efficiently learn. This challenge seems poorly investigated and doesn't result in a clear answer.
- The paper has a heavy focus on results from the synthetic task CMSP. Additionally, they do not look at the effect of their proposed methods on non-hierarchical baselines. The mention two other synthetic tasks that they drew inspiration from which they could have added to their evaluation. While you could argue that the papers that originally introduced these benchmarks already explored the effects of learning these synthetic tasks, it is not clear in the paper that the aim of these investigations was to investigate the learning of narrow tasks which is what this work wishes to explore. The two non-synthetic experiments, MNIST and LLMs on Python documentation, are only briefly mentioned and only offer that pruning works better than distillation without much insight into these findings. Additionally, the group lasso regularisation is not used on the LLM experiment.
- The paper has a few areas where a speculative reason is provided to explain some phenomena and is then later used as the grounding for a future speculative explanation. For example "One explanation of these results is that networks compute the parity of the atomic subtask bits in the first hidden layer, and then to learn the composite subtask they compute the parity of these values in the second layer." However, this mechanism is never empirically validated. No activation probing, representational analysis, or layer-specific ablation is performed to support this claim. Instead, a comparison between shallow and deeper networks is offered, but this is insufficient to rule out the impact of increased expressivity due to added depth. While a width-matched baseline is included in the appendix, it does not eliminate the core confounding point that deeper networks may simply represent more complex functions by design. This statement is later used to add legitimacy to the broader claim "This result may also provide some explanation for advantage of depth in deep learning – not only is depth necessary for networks to efficiently approximate certain functions [33], we find here that depth is helpful to efficiently learn tasks with hierarchical structure." The use of one unproven explanation as evidence for another raises concerns about the paper’s reasoning rigor in the discussion section.
- The paper could greatly benefit from another look at the figures and their captions. For example figure 3 has visualizations of 2-hidden-layer MLPs to provide evidence of the use of group lasso regularisation however an explanation for the lines is never given (what do the colours represent? Why is the line thickness different? Is this weight magnitude? Activation size?). I find figure 4 left difficult to parse as the authors have a three dimensional result which they are showing in a 2D graph so gleaning any real patterns requires a lot of thought on the readers side. Figure 5 right has shadow lines but no explanation is given for what this is. It cannot be an STD or error estimate because it is offset from the main lines in some cases and I can't even guess what these shadow lines mean.
- The limitations section of this paper is poor and lacks an honest self-critique. The authors point out they only look at one, simple pruning strategy and lack of LLM hyperparameter tuning as the only limitations in their work. They then have another paragraph that raises philosophical challenges related to narrow AI and postulate that this entire direction of work could be irrelevant if neural networks learn in a way that would negate this subfield of research. This is not a useful limitations section and ignores some real limitations of the presented work.

---

> ### Author Rebuttal · Authors · 2025-07-31
>
> Thank you for this detailed and thoughtful review! We’re so glad that you found the paper’s topic to be interesting and important, that you found the CMSP task to be useful, and that you found the paper to be laid out well! We appreciate the opportunity to improve both the results and the clarity the paper based on your comments:
>
> > The paper claims to explore when it is possible to train narrow models from scratch and concludes that "Our work shows that for some types of tasks, it may be necessary to train on a broad data distribution in order to learn." The authors only investigated one type of task structure - highly hierarchical - and found you needed a broad dataset to efficiently learn. This challenge seems poorly investigated and doesn't result in a clear answer.
>
> We think that our results on the hierarchical task, CMSP, are impactful since they are a proof-of-concept that extremely strong curriculum learning effects can exist when the data has a particular structure. We think that this task provides a very nice “model organism” for studying curriculum learning and nonlocal representations that will be of independent interest to the theory community. We do not claim that all realistic tasks display this structure; rather, we use CMSP as an existence proof that some tasks are necessarily hierarchical. We thank the reviewer for pointing this out and plan to revise the paper to more clearly articulate this point.
>
> > The paper has a heavy focus on results from the synthetic task CMSP. Additionally, they do not look at the effect of their proposed methods on non-hierarchical baselines... Additionally, the group lasso regularisation is not used on the LLM experiment.
>
> Ah, thanks for this recommendation! To provide a clear baseline, It would have been appropriate for us to also study curriculum learning and our group lasso regularization method on the simpler MSP task, as well as on CMSP, and we could include results on this in the final version if you’d like! We’d guess that MSP won’t have very meaningful interactions between the subtasks (since the relevant task bits are mostly distinct between subtasks), but it is worth checking. We also note that if there were meaningful interactions between subtasks on baselines like MSP, this would demonstrate that these tasks are also hierarchical, rather than invalidating the curriculum learning dynamics seen on CMSP.
>
> We’re also sorry that our discussion of our results on real-world tasks (MNIST and LLMs) was not clear. On MNIST and LLMs, we did meaningfully study the creation of narrow networks with the same methods we used on CMSP -- pruning without and without group sparsity regularization, distillation, and training from scratch. You said that we do not use group lasso regularization on the LLM experiments, but Figure 5 shows the results of using group sparsity regularization to prune LLM MLP neurons. We will aim to improve the presentation of these results in the final version of the paper -- Figure 5 is indeed a complicated figure, and its caption could be improved.
>
>
> > The paper has a few areas where a speculative reason is provided to explain some phenomena and is then later used as the grounding for a future speculative explanation...
>
> Thank you for the opportunity to improve the rigor of this section. In the sentence “One explanation of these results”, we meant to say “One potential explanation of these results could be…”. If you’d like, we could also perform activation probing in the discussion period to test this claim -- this is a great suggestion. Also, the sentence “This result may also provide some explanation for advantage of depth in deep learning – not only is depth necessary for networks to efficiently approximate certain functions [33], we find here that depth is helpful to efficiently learn tasks with hierarchical structure” is based on our empirical result that deeper, parameter-matched networks learn CMSP faster on average than shallow networks. We don’t think it depends on our earlier speculation being true that networks were solving the task in a particular manner. However, we do think it would be valuable to investigate whether this increased training efficiency is a direct result of the hierarchical structure—for instance, we could see if this same effect is present on non-hierarchical baseline tasks, if you would find this helpful. We thank you for suggesting that we further disambiguate these possible explanations.
>
> > The paper could greatly benefit from another look at the figures and their captions. For example figure 3 has visualizations of 2-hidden-layer MLPs to provide evidence of the use of group lasso regularisation however an explanation for the lines is never given (what do the colours represent? Why is the line thickness different? Is this weight magnitude? Activation size?). I find figure 4 left difficult to parse as the authors have a three dimensional result which they are showing in a 2D graph so gleaning any real patterns requires a lot of thought on the readers side. Figure 5 right has shadow lines but no explanation is given for what this is. It cannot be an STD or error estimate because it is offset from the main lines in some cases and I can't even guess what these shadow lines mean.
>
> Thank you for drawing our attention to this problem! We’d be happy to clarify this in the final version of the paper. To answer your questions briefly:
> In Figure 3, the colours represent whether the weights were positive or negative, however that doesn’t influence the takeaway of that figure, which is just that the regularized network is far sparser than the unregularized network. The transparency “alpha” parameter of these lines as well as their width is set based on the magnitude of the weight, which is all we really care about here. Thank you for pointing out that this is not clear enough in the current figure and plan to address it in the caption for the final version.
> In Figure 5, the shadow lines show the loss over training on each batch, while the primary lines show an exponential moving average of the losses. We will also clarify this in the caption.
>
> > The limitations section of this paper is poor and lacks an honest self-critique. The authors point out they only look at one, simple pruning strategy and lack of LLM hyperparameter tuning as the only limitations in their work. They then have another paragraph that raises philosophical challenges related to narrow AI and postulate that this entire direction of work could be irrelevant if neural networks learn in a way that would negate this subfield of research. This is not a useful limitations section and ignores some real limitations of the presented work.
>
> Thank you for pointing this out. We will substantially expand the limitations section in the final version of the paper. Part of the reason why the limitations section is currently short is simply that we were running up into the page limit. In our final version, we will include some discussion about the limited task diversity in our synthetic study and the fact that parts of our analysis are done only on synthetic tasks. Note however that we actually did apply group lasso training to LLMs, as discussed above.
>
> Also, the more philosophical final paragraph is not really part of the limitations; this will be more clear in the final version of the paper.
>
> > Have you done any exploration into tasks that have a hierarchical structure in the presentation of the data but not necessarily the task? For example the use of abstractions in image based tasks, where there is an underlying structure that can make the task easier but is not directly tied to the result.
> We have not! That could be interesting in future work!
>
> > By removing the ability to recover performance on a different task, is it possible to degrade performance on your current task if they are closely linked? This seems to me to be a bit of a random proxy to be optimizing for that could possibly lead to degradation of the whole system if done in some settings.
>
> When doing group sparsity regularization, the network is being trained to maintain its performance on a given task and so its performance typically does not degrade unless one regularizes too strongly. Note that the group lasso sparsity penalty is applied in conjunction with a standard data loss term (eg, cross entropy loss on the relevant task).
>
> > As the paper stands, I would raise my score if the figures are better labelled and explained, the language and discussion was made a bit more rigorous and if the limitations section was a more critical reflection of the work actually done in this paper.
>
> Thank you for the thoughtful engagement with our results and the opportunity to raise our score by making our figures clearer and limitations more rigorous. While we can’t update the manuscript in the discussion period, we believe that it would be quite straightforward to make these improvements to the clarity and rigor of the work that we’ve discussed above. If necessary, we could draft these improvements in the discussion period and present them in text form (e.g. for the new limitations section and some captions) if you’d like? Either way, we agree that the figures could be clearer and the limitations could be more rigorous, and we will be sure to include the aforementioned changes in the final version of the paper.

---

> > ### Comment · Reviewer_gjJ8 · 2025-07-31
> > **Reponse**
> >
> > Thank you for your rebuttal.
> >
> > As I said previously the main issues I had with the paper were presentation based. Given the commitment to improve this I will change my score. I think this is an interesting piece of work that could lead to some interesting future directions. Also thank you for the clarification on the group lasso regularisation on the LLM experiment.

---

### Official Review · Reviewer_yXiG · 2025-07-02

**Clarity:** 3
**Significance:** 2
**Originality:** 2
**Rating:** 4
**Confidence:** 4

**Summary:**

The main question that this paper aims to answer is:

what if we try to create a specialized ("narrow") model by extracting specific skills from a more general-purpose larger model? Which are the main challenges in doing so?

The authors focus on two specific challenges:
1) some narrow tasks will require broad training curricula because of hierarchical dependencies between tasks. They explore this issue using a synthetic task called CMSP (compositional multitask sparse parity -- it is a variation of a task that another proposed earlier).
2) nonlocality of representations: the problem here is it is hard to isolate specific skills through direct pruning because of the distributed way in which neural networks learn representations.

The paper presents some structured pruning methods (that are not new themselves -- e.g., group-lasso regularization) to extract and transfer narrow skill subsets into smaller networks.

**Questions:**

* I suggest that the authors read the following paper:
https://arxiv.org/pdf/2406.08658  (Pruning is Optimal for Learning Sparse Features in High-Dimensions)
It is a more theoretical paper but I think it is very relevant.

* One of the paper's main claims is that the  distributed nature of representations is a significant challenge for pruning. However the pruning can be done during training or fine-tuning for that more narrow/specific task. Would such an approach challenge this main claim?

* The paper provides some limited results for experiments involving LLMs (Llama), evaluating on Python token prediction. I admit that I found that part of the paper rather speculative and not very convincing. For example, the authors state about Fig-6 "we tentatively find that pruning substantially outperforms training from scratch and distilling a model from scratch on the data-parameter frontier" -- or later in the same section:
"in our case it does not seem like the relevant features for our task are localized into a set of “Python” vs.“non-Python” neurons, at least that attribution pruning identifies."
It is important to have such experiments because they bring the paper closer to the "real world" -- but it is also necessary for these experiments to be done more deeply/extensively and the analysis of the results to be more conclusive.

**Ethical Concerns:**

["NO or VERY MINOR ethics concerns only"]

**Final Justification:**

I have increased my score from borderline reject to borderline accept after reading the rebuttal and the more positive review of the paper by reviewer gjJ8.

**Limitations:**

yes

**Quality:**

3

**Strengths And Weaknesses:**

Main strengths
* The topic (at a high level) is significant because extracting narrow, specialized AI models (through pruning, distillation, etc) efficiently from general-purpose models is a practically important area for scaling up AI systems.
* The paper’s experimental setup is clearly described. Figures and visualizations are informative and well done.
* The writing is also clear and good.

Main Weaknesses
* In my opinion, the paper does not have a very clear result or "message". It reads more as an "exploration report",  showing some challenges in extracting narrow models from more general-purpose models -- but it is not clear at the end of the day whether these challenges would still exist if the "extraction process" was different. The conclusions of the paper are specific to the specific synthetic hierarchical task that is considered and the specific pruning methods that are considered.
* There is a lack of theoretical or clear analytical results explaining why and how hierarchical curriculum learning and non-locality arise from neural network training. The results rely on empirical observations.
* The main insights (hierarchical curriculum learning, nonlocality, the use of structured pruning methods) do not come across as highly original. These are well understood concepts in the broader neural network literature.

---

> ### Author Rebuttal · Authors · 2025-07-31
>
> Thank you for your comments and suggestions! We’re glad you found the paper’s topic significant and practically important, and that you found its experiments, figures, and writing clear and informative. We’re grateful for this opportunity to clarify some points and improve the paper in response to your comments and questions:
>
> > In my opinion, the paper does not have a very clear result or "message". It reads more as an "exploration report", showing some challenges in extracting narrow models from more general-purpose models -- but it is not clear at the end of the day whether these challenges would still exist if the "extraction process" was different. The conclusions of the paper are specific to the specific synthetic hierarchical task that is considered and the specific pruning methods that are considered.
>
> We agree that the paper has more of an exploratory structure than the average machine learning paper. However, we think that this exploratory structure is appropriate given that the paper focuses on understanding some more conceptual questions about how networks learn and implement their skills that are broadly relevant to understanding the performance of many different methods. The paper makes connections between subfields (like learning dynamics, mechanistic interpretability, pruning, and unlearning) and lays a conceptual groundwork for future work. While this exploratory tone is somewhat nonstandard, we think the paper will still be of interest to folks from multiple communities. We also think that many of our specific results are interesting and novel, including (1) the introduction of the CMSP task as a model organism for understanding curriculum learning, distributed representations, and unlearning, (2) the comparison of which methods are most efficient for creating narrow models, and (3) the use of sparsity regularization training for unlearning. We appreciate the reviewer raising this concern, and we will aim to better emphasize and contextualize these contributions in the final version of the paper.
>
> > There is a lack of theoretical or clear analytical results explaining why and how hierarchical curriculum learning and non-locality arise from neural network training. The results rely on empirical observations.
>
> We note that there is already a strong theoretical basis for nonlocality arising from neural network training, which we discuss in detail in lines 172-183. While we agree that a mathematical formalism for curriculum learning would be useful, we think that there is substantial value in introducing CSMP and studying it empirically as a first step. For instance, the original exploratory study of the MSP task in Michaud et al. (NeurIPS 2023) then inspired the theoretical work of Fonseca et al. (NeurIPS 2024) and together these helped inspire Ren at al (2025). We are hopeful that the CMSP task we introduce here could inspire similar follow-up efforts in the theory community towards understanding curriculum learning, and that this possibility will make the paper of interest to the theory community regardless.
>
> The main insights (hierarchical curriculum learning, nonlocality, the use of structured pruning methods) do not come across as highly original. These are well understood concepts in the broader neural network literature.
>
> We think there is value in drawing connections between these insights, and that many of our specific contributions do provide significant novelty. For instance, the CMSP task is novel, and while sparsity regularization had been used in prior work to help with structured pruning, it had not been studied for its use in unlearning, and prior work had not analyzed whether it is efficient for creating narrow AI systems (which is a significantly different task from standard pruning). We will do a better job of clarifying these contributions in the final paper.
>
> > I suggest that the authors read the following paper: https://arxiv.org/pdf/2406.08658 (Pruning is Optimal for Learning Sparse Features in High-Dimensions) It is a more theoretical paper but I think it is very relevant.
>
> Thanks so much for this suggestion! This work is very relevant, and we’ll be sure to cite it in the final paper.
>
>
> > One of the paper's main claims is that the distributed nature of representations is a significant challenge for pruning. However the pruning can be done during training or fine-tuning for that more narrow/specific task. Would such an approach challenge this main claim?
>
> We expect that the distributed nature of representations will be relevant whenever one does pruning. Is this question specifically asking about the approach of slowly pruning model components without regularization while fine-tuning on a narrow task? If I recall correctly, we did some experiments on this method several months ago, and while it works somewhat, it was pretty sensitive to hyperparameters and not obviously better than first pruning and then finetuning. If you’d like we could revisit these results and discuss them in the discussion period? Either way, we don’t think it challenges the claim that distributed representations make pruning more difficult than it otherwise would be (if one could prune concept-specific neurons without needing to alter the network’s representations at all).
>
> > The paper provides some limited results for experiments involving LLMs (Llama), evaluating on Python token prediction. I admit that I found that part of the paper rather speculative and not very convincing. For example, the authors state about Fig-6 "we tentatively find that pruning substantially outperforms training from scratch and distilling a model from scratch on the data-parameter frontier" -- or later in the same section: "in our case it does not seem like the relevant features for our task are localized into a set of “Python” vs.“non-Python” neurons, at least that attribution pruning identifies." It is important to have such experiments because they bring the paper closer to the "real world" -- but it is also necessary for these experiments to be done more deeply/extensively and the analysis of the results to be more conclusive.
>
> While we were humble in our presentation of these results, we do think that they are informative and impactful in their current form. We think the ‘”Python” vs. “non-Python”’ neuron pruning results are already pretty conclusive, given that the attribution method tracks the true importance of the neurons pretty well (Figure 11, left). In the discussion period, we could provide some further evidence by doing complete ablations of the neurons within a layer and showing that the distribution over effect sizes on Python documents is fairly uniform, if that would be helpful.
>
>
>
> References:
>
> Michaud, Eric, et al. "The quantization model of neural scaling." Advances in Neural Information Processing Systems 36 (2023): 28699-28722.
>
> Fonseca, Nayara, et al. "An exactly solvable model for emergence and scaling laws in the multitask sparse parity problem." Advances in Neural Information Processing Systems 37 (2024): 39632-39693.
>
> Ren, Yunwei, et al. "Emergence and scaling laws in sgd learning of shallow neural networks." arXiv preprint arXiv:2504.19983 (2025).

---

> > ### Comment · Reviewer_yXiG · 2025-08-02
> >
> > Thank you for this very well written rebuttal. I appreciate the clarity and honesty of your responses.
> >
> > I also appreciate the other reviews -- especially the review by gjJ8 -- which helped me look at the paper a bit more positively. I will increase my score to the next level (borderline accept).
> >
> > However my main concern that this work is a bit premature -- without a clear message/conclusion -- still remains. I am certain that if you work on this paper a bit more, considering these constructive reviews you have received from NeurIPS, you will have a much stronger publication at the end.

---

### Official Review · Reviewer_kBj4 · 2025-07-02

**Clarity:** 3
**Significance:** 3
**Originality:** 4
**Rating:** 4
**Confidence:** 3

**Summary:**

This paper studies the challenges in creating _narrow_ AI: "tool" AI systems designed to have specific skills as opposed to general-purpose models with broad skills. This is motivated from the viewpoints of information efficiency as well as preservation of human agency. The study is conducted from two broad angles: (1) the efficacy of training a narrowly-skilled model from scratch, and (2) how best one could transfer specific skills (while discarding the rest) from a generalist model into a specialist model.

Regarding (1), an empirical study using the compositional multitask sparse parity (CMSP) task is conducted. The task is designed such that certain difficult subtasks are "composites" of easier, "atomic" subtasks, inducing the possibility for "curriculum learning". It is observed that neural networks indeed learn the compositional task much faster when also trained on the atomic tasks, while training from scratch only on the compositional "narrow" subset is sample-inefficient (i.e., requires more samples in total to reach the same performance) -- this indicates that those easier skills cannot be safely skipped when training from scratch. They also find that depth helps learning to solve CMSP, leading to the hypothesis that many deep networks indeed learn hierarchical tasks in the same staggered fashion.

Regarding (2), pruning and distillation are explored as candidate methods for selectively transferring skills from a broader model into a smaller model. This is analysed using the CMSP task above, MNIST digit classification, as well as LLM next-token prediction. They find that the circuitries for different tasks are entangled: this makes pruning for specific skills a non-trivial task. To address this problem, they propose a group lasso regularization that aims to "clean up" entire groups of neurons that are irrelevant to the narrow task at hand. The observations suggest that pruning the model whilst applying the proposed regularization, is more efficient in comparison to distillation.

**Questions:**

Do the authors have any ideas/plans for extending the experimental setup in §3 and §4 to more realistic datasets?

**Ethical Concerns:**

["NO or VERY MINOR ethics concerns only"]

**Final Justification:**

My main concern was that the paper relies too much on toy experiments, and that empirical breadth is not sufficiently covered. In addition to this, I have realised after reading the other reviews that there remains a lack of a clear "message" and that the paper is at times a bit disorganized in presenting its main ideas. These reasons prevent me from increasing the final score to 5 at this point. At the same time, I believe the paper already has sufficient merit to be considered for acceptance, hence, my recommendation remains at Borderline Acceptance.

**Limitations:**

The authors mention limitations in §7, but should mention that CMSP is a toy task (which they do mention in lines 159-162 but should also briefly include in §7 for completeness).

**Paper Formatting Concerns:**

I have no formatting concerns.

**Quality:**

4

**Strengths And Weaknesses:**

### Strengths

1. The questions explored are insightful, interesting and well-motivated. With general-purpose MLLMs taking over the world, this kind of work that explores specialist models instead, is well-timed.
1. The proposed compositional multitask sparse parity task and the group lasso regularization for pruning can, in themselves, be valuable for future work. Notably, pruning with the sparsity regularization seems to be an effective strategy for narrow skill transfer.
1. The paper is fairly well-organized given the density of information covered. Sections follow each other logically.

### Weaknesses

1. The curriculum learning claims are based entirely on the toy case study of compositional multitask sparse parity.  While these experiments themselves are quite thorough, they do not guarantee transfer to real-world cases: the authors note this in §3.2. Limitations noted in §7 also apply.
1. The authors conduct LLM experiments in §5, but in contrast, the vision counterpart of these experiments in §4 is confined to MNIST. Given that vision is just as important as language in modern AI systems, I would be curious to see whether the results transfer to more realistic vision datasets -- absence of these results does impact the paper's significance.

Overall, I believe the strengths sufficiently outweigh the weaknesses: enough for me to recommend borderline acceptance at this stage. I would have considered a higher score if more realistic-dataset experiments had been presented, but I would be happy to reconsider my score during the discussion phase and after looking at insights from other reviewers.

---

> ### Author Rebuttal · Authors · 2025-07-31
>
> Thank you for your thoughtful summary and review! We’re so glad you found the paper insightful, interesting, well-motivated, well-timed, and well-organized. We’ll comment on the weaknesses you identified below:
>
> > The curriculum learning claims are based entirely on the toy case study of compositional multitask sparse parity. While these experiments themselves are quite thorough, they do not guarantee transfer to real-world cases: the authors note this in §3.2. Limitations noted in §7 also apply.
>
> We agree that the compositional multitask sparse parity task and the group lasso regularization method would be worth studying in more detail in future work, and that our work could be improved by including additional evidence of realistic curriculum learning beyond the toy setting of CMSP. However, we want to emphasize that our CMSP experiments alone will be of substantial interest to the theory community: they provide an extremely simple model organism for studying curriculum learning and distributed representations. For instance, the original exploratory study of the MSP task in Michaud et al. (NeurIPS 2023) then inspired the theoretical work of Fonseca et al. (NeurIPS 2024) and together these helped inspire Ren at al (2025). We are hopeful that the CMSP task we introduce here could inspire similar follow-up efforts in the theory community towards understanding curriculum learning, and that this possibility may make the paper of interest to a particular audience at the conference. We also note that prior work has studied hierarchical learning dynamics in more realistic settings, which we mention on lines 40-41 and 302-303, providing a stronger foundation for the claim that curriculum learning is relevant to real-world cases.
>
> > The authors conduct LLM experiments in §5, but in contrast, the vision counterpart of these experiments in §4 is confined to MNIST. Given that vision is just as important as language in modern AI systems, I would be curious to see whether the results transfer to more realistic vision datasets -- absence of these results does impact the paper's significance.
>
> We agree that it would strengthen the paper to include more comprehensive vision model experiments and we wish that we had had the time and space to include these! While we didn’t have time to run them in the rebuttal period, we may have time to run them in the discussion period. Do you have any recommendations? One option would be to run experiments similar to our MNIST results but on CIFAR-10 and CIFAR-100.
>
> Another important observation in the paper was that the group lasso regularization was very effective for targeted unlearning of CMSP tasks. To expand on this, we could extend our LLM and vision experiments to analyze whether group lasso regularization is also an effective unlearning method in these more realistic settings. For instance, we could fine-tune our Python-specific LLMs on a variety of other tasks and compare the networks pruned with group lasso training to more naive pruning baselines. We could also run these experiments in the discussion phase if they would be of interest?
>
> References:
>
> Michaud, Eric, et al. "The quantization model of neural scaling." Advances in Neural Information Processing Systems 36 (2023): 28699-28722.
>
> Fonseca, Nayara, et al. "An exactly solvable model for emergence and scaling laws in the multitask sparse parity problem." Advances in Neural Information Processing Systems 37 (2024): 39632-39693.
>
> Ren, Yunwei, et al. "Emergence and scaling laws in sgd learning of shallow neural networks." arXiv preprint arXiv:2504.19983 (2025).

---

> > ### Comment · Reviewer_kBj4 · 2025-08-03
> > **Response to authors' rebuttal**
> >
> > Thank you for this thoughtful rebuttal. I believe that out of the proposed ideas (quoting the authors)
> >
> > > run experiments similar to our MNIST results but on CIFAR-10 and CIFAR-100
> >
> > and
> >
> > > fine-tune our Python-specific LLMs on a variety of other tasks and compare the networks pruned with group lasso training to more naive pruning baselines
> >
> > I'd very much like to see the second experiment in a refined version of the paper. This would perhaps also address Reviewer yXiG's concern laid out in Q3 : perhaps with these experiments you can show more instances of validation towards your claim regarding skill localisation ("Python" vs "non-Python" -- lines 277-279). If you wish to add vision experiments, I'd suggest scaling up to ImageNet-like datasets as CIFAR-{10, 100} is, in my very personal opinion, not very different from MNIST in terms of scale.
> >
> > Although I found the paper fairly well-organized earlier, I do now resonate with Reviewers tPS8 and yXiG that the paper is a bit premature in terms of presenting a clear message or conclusion, and that there is indeed a disconnect between the two main parts (learning of hierarchical skills / pruning vs distillation). As such, I intend to maintain my current rating for now and hope that, in the event that the paper does not make it into NeurIPS, the authors continue to refine it using the suggestions presented here.

---

### Note · Authors · 2025-08-16

First, we'd like to thank the reviewers for a thoughtful and candid discussion about our paper. Overall, a pretty clear consensus has emerged. The reviewers found the paper to be “insightful, interesting and well-motivated", “well-timed", that its "experiments are well motivated", and that “the writing is also clear and good." After the rebuttal stage, 3/4 reviewers now say they recommend acceptance, with one recommending borderline rejection.

Overall, we think the AC should favor acceptance for the following reasons:

* We still think that the reviewers are underrating the potential impact of the synthetic task (CMSP) introduced in the paper. We are excited about CMSP since it is a very clean generalization of the "multitask sparse parity" (MSP) task introduced previously in the literature, which went on to inspire many works including Fonseca et al. (NeurIPS 2024) and Ren et al. (2025). CMSP has a property which MSP (and most other tasks) do not: neural networks trained on CMSP exhibit extremely dramatic curriculum learning effects. We think that this "model organism" of curriculum learning will be of interest to a variety of communities, notably the deep learning theory community.
* Our comparisons of methods for ``narrowing'' models (pruning with group lasso regularization vs attribution pruning vs and distillation) are novel and noteworthy.
* Multiple reviewers said that there is a clear path to further improving the paper. Reviewers suggested specific changes or experiments which can be included in our final work while raising their score and expressing confidence in our ability to improve the paper.

While the reviewers feel that the paper could be made more complete (and we agree!), we think that the paper deserves to be accepted because it presents an earnest exploration of a new and important topic: the study of methods for efficiently creating narrow AI systems from larger general ones. The paper doesn't have any particularly shocking results, and we are quite candid about this fact (reviewer tPS8: “I especially appreciate that the authors don’t embellish their results"). Nevertheless, we think there is much in the paper which will be of interest to various communities, and that we should not let the perfect be the enemy of the good.

---

### Decision · Program_Chairs · 2025-09-17

**Decision:**

Accept (poster)

**Comment:**

This paper explores the challenges of creating small, specialized AI models, finding that some narrow skills paradoxically require training on broad data due to hierarchical learning dependencies. When transferring capabilities from large to small models, the authors show that pruning can outperform distillation even though skills are not neatly localized, and they investigate using regularization to better align skills with prunable components. The paper presents experiments on a synthetic task (compositional multitask sparse parity), MNIST, and language modeling on Python documents.

This is a well-written, creative paper, and the experiments including the novel synthetic task are well-chosen. It frames known techniques such as pruning and distillation in an original problem setting that has the potential to inspire interesting future work. Multiple reviewers noted that the study was still somewhat inconclusive, but the consensus among reviewers and AC was that there were enough interesting findings to merit publication.